# Morphological Study of Tetra-n-Butylammonium Bromide Semi-Clathrate Hydrate in Confined Space

## Lijuan Gu [1,2] and Hailong Lu [1,2,*]

1   Beijing International Center for Gas Hydrate, School of Earth and Space Sciences, Peking University, Beijing 100871, China; glj0317@pku.edu.cn
2   Technology Innovation Center for Carbon Sequestration and Geological Energy Storage, MNR, Beijing 100091, China
*   Correspondence: hlu@pku.edu.cn

**Abstract:** Tetra-n-butylammonium Bromide (TBAB) finds extensive use in diverse applications. An in-depth investigation into the effects of the formation conditions on TBAB hydrate is necessary to optimize the application process. This work focuses on examining the influence of the mass concentration of TBAB solution and the cooling rate on TBAB hydrate formation through optical microscopy and Raman spectroscopy. The TBAB hydrate formation process occurs in a confined space created by an optical sheet with a 0.03 mm deep groove. Four TBAB solutions of 13. 8 wt%, 18 wt%, 32 wt%, and 40 wt% are investigated, and the supercooling required for hydrate nucleation increases with concentration at a cooling rate of 0.5 K/min. Notably, Type A TBAB hydrate preferentially forms in all of the solutions, although type B hydrate is thermodynamically stable in the two dilute solutions. At a larger cooling rate of 2 K/min, two distinct crystal growth patterns are observed: one controlled by mass transfer and the other regulated by heat transfer. Increasing the cooling rate not only alters the optical morphology, but also reduces the supercooling due to a decrease in the Gibbs free energy barrier caused by a larger temperature gradient. This is beneficial for practical applications as it helps to alleviate the supercooling degree.

**Keywords:** Tetra-n-butylammonium Bromide; optical morphology; crystal growth; supercooling

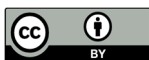

## 1. Introduction

Semi-clathrate hydrates (SCHs) have extensive applications in various fields [1], including carbon dioxide capture and separation (CCS) [2–4], hydrogen storage [5], and air conditioning systems [6]. In semi-clathrate hydrates, cations occupy the voids of the framework and ionically interact with the water–ionic polyhedral framework, allowing them to remain stable above the freezing point under ambient pressure. Among all the semi-clathrate hydrates, TBAB has been the subject of extensive research.

In TBAB solutions, TBAB polyhydrates can be formed under different conditions, including the orthorhombic TBAB·38H$_2$O [7], three tetragonal hydrates with different stoichiometry (24, 26, and 32) [8,9], and a trigonal hydrate with small hydration number of $2\frac{1}{3}$ [10]. Among all the structures, TBAB·26H$_2$O and TBAB·38H$_2$O are two of the most stable structures. They are also named as type A and type B TBAB hydrates [11]. The detailed structure of TBAB·38H$_2$O has been determined, in which the host framework is constructed by Br atoms and water molecules through hydrogen bonds and the TBA cations are located at the center of two tetrakaidecahedrons and two pentakaidecahedra [7]. The detailed structure of the tetragonal type A hydrate has not been resolved due to the existence of several metastable phases [9].

Morphology studies deal with the observation of TBAB hydrate nucleation, the growth of hydrate crystals, and the characteristic appearance, including the size, shape, and growth pattern of crystals, through a microscope [12,13]. Morphological observations

are very useful in understanding the dynamics of TBAB hydrate crystal growth to optimize its industrial application process [14].

Morphological studies on TBAB SCH and mixed gas/TBAB hydrate have been conducted in the literature [13,15–17]. The free growth of TBAB SCH was observed with an optical microscope at different supercooling levels [15]. Single crystals of TBAB SCH were observed at the tip of a wire chilled with liquid nitrogen in a thin glass capillary. Ye et al. investigated the crystal morphology of bulk TBAB hydrates with an optical microscope [18]. Shimada et al. examined the recrystallization probability of type A TBAB hydrate at various mass concentrations of aqueous solution with an optical microscope [19]. However, these explorations primarily focused on the growth of bulk TBAB SCH crystals, making it challenging to capture and quantify the dynamic process of hydrate formation. Additionally, the influence of the formation conditions on TBAB SCH have not been fully understood, which is crucial in its practical application. Among the various factors that influence crystal growth, the mass concentration of TBAB solution and the cooling rate should be investigated in depth.

In this study, we carry out in situ morphology observation in a confined space using a polarizing microscope. The confined space is made with an optical sheet etched with a groove of 0.03 mm depth. The optical sheet is laid on a reactor capable of withstanding high pressure up to 5 MPa and low temperature down to minus 110 °C. The dynamical TBAB SCH formation process from TBAB solutions with different mass concentrations is recorded under large supercooling. The type of TBAB crystal formed in solution with different mass concentrations is identified with confocal laser Raman spectroscopy. Furthermore, the influence of the cooling rate on hydrate morphology is investigated.

## 2. Materials and Methods

The experimental setup is shown in Figure 1, which consists of a high-pressure/low-temperature reactor, a polarizing microscope, and a self-designed optical sheet for in situ SCH formation and observation. The reactor has a maximum operating pressure of 5 MPa and a minimum temperature of −110 °C. The reactor is equipped with a pressure transducer and a temperature sensor, Pt-100, providing pressure and temperature stability within ±1 kPa and ±0.1 °C, respectively. The optical window of the reactor is made of calcium fluoride glass. Temperature control is achieved with circulating liquid nitrogen and an electric heating wire wrapped around the sample holder inside the reactor. The liquid nitrogen bath, the electric heating component, and the polarizing microscope are all connected to a personal computer for online operation.

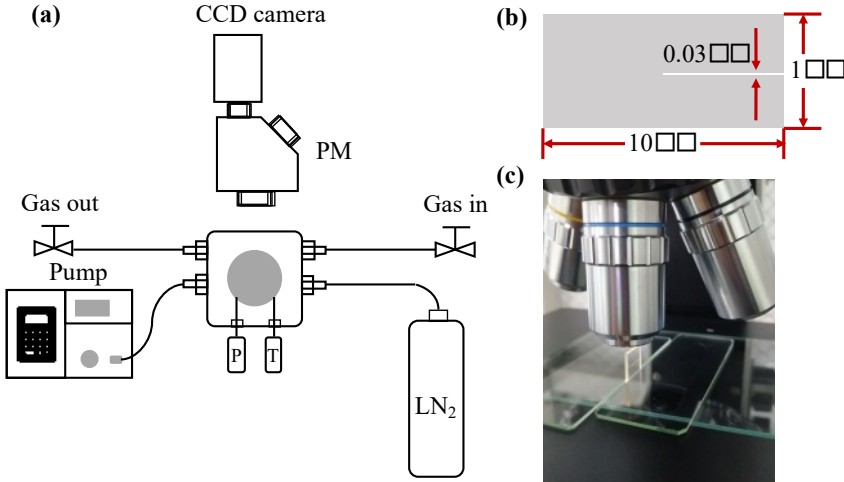

**Figure 1.** (**a**) Schematic diagram of the experimental setup. PM: Polarized microscope; LN₂: liquid nitrogen tank. (**b**) Dimension of the optical sheet. (**c**) Picture of the optical sheet examined with the optical microscope.

The deionized water used in the experiments was from Meilunbio. The TBAB used in this work was of high purity (99%) and supplied by Bioruler. Four TBAB solutions with mass fractions of 13.8 wt%, 18 wt%, 32 wt%, and 40 wt% were prepared for the experiments. The formation process of the TBAB semi-clathrate hydrate was observed using a polarizing microscope (Shanghai Huitong Optical Instrument Co., LTD, Shanghai, China, HPL-30C). To examine the optical properties of the samples with the polarizing microscope, we designed an optical sheet made of silica glass with dimensions of 10 mm × 5 mm × 1 mm. Within this sheet, an optical groove with dimensions of 10 mm × 5 mm × 0.03 mm was created. The TBAB solution was injected into the groove using a 1 mL syringe.

A detailed description of the experimental setup for in situ Raman spectroscopy can be found in a previous work [20]. A confocal Raman imaging microscope (WITec Inc., Ulm, Germany, alpha300R) was adopted. It operates with a laser wavelength of 532.1 nm and a power of 10 mW. The Raman spectra range from 100 to 4000 cm⁻¹. The reactor and its temperature controlling system is the same as that in Figure 1.

### 3. Results

*3.1. Hydrate Nucleation at Low Cooling Rate*

3.1.1. Supercooling Required for Hydrate Nucleation

Large degree of supercooling is required for the nucleation of the TBAB SCH [21]. We conducted multiple TBAB SCH formation experiments using each solution. The temperature of the reactor gradually decreased from 13 °C to −10 °C at a cooling rate of 0.5 °C/min. Hydrate formation occurs during the cooling process, and the nucleation temperature of the TBAB solution with different mass concentrations was measured and is shown in Figure 2. As shown in Figure 2, nine rounds of experiments were conducted for each TBAB solution and the nucleation temperature exhibited stochastic behavior. The average TBAB SCH formation temperature was found to be −6.2 °C, −6 °C, −6 °C, and −5.8 °C for the 13.8 wt%, 18 wt%, 32 wt%, and 40 wt% solutions, respectively. Two types of TBAB hydrates can grow in aqueous solution, while the equilibrium temperature of type A and type B TBAB hydrates in solutions of different concentrations are different. In order to determine the degree of supercooling that is required for hydrate nucleation, the hydrate type should be firstly identified.

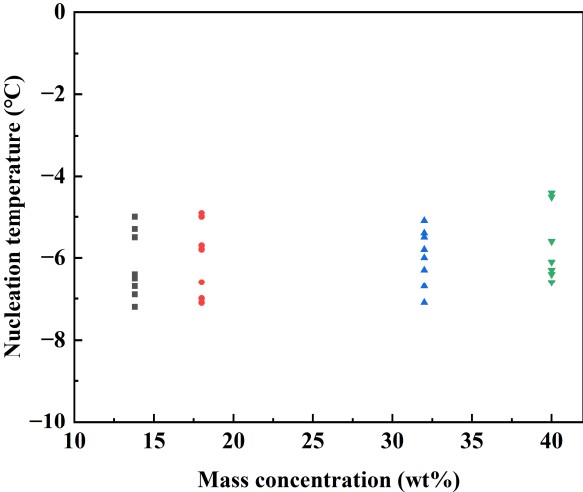

**Figure 2.** The nucleation temperature of 13.8 wt%, 18 wt%, 32 wt%, and 40 wt% TBAB solution at a cooling rate of 0.5 °C/min.

The Raman spectra of the TBAB SCH, formed at different TBAB solution concentrations, are presented in Figure 3. The spectral range between 2800 and 3600 cm⁻¹ was investigated, which encompasses the fingerprint peaks of different types of TBAB SCH

[16,22]. The crystal structure of the type B TBAB hydrate have been successfully determined, in which the tetra-n-butylammonium cation is located at the center of four cages, viz. two tetrakaidecahedra and two pentakaidecahedra [7]. Although the detailed structure of type A TBAB hydrates has not been identified [9], it is suggested that the cation is located at the center of three tetrakaidecahedra and one pentakaidecahedron [11]. Thus, the spectral bands of the C-H stretching modes of the butyl groups that correspond to a Raman shift of 2750–3150 cm⁻¹ [22] show differences between type A and type B hydrates. The type A TBAB hydrate shows three major peaks at 2881, 2919, and 2939 cm⁻¹, as well as one broad band between 2966 and 2980 cm⁻¹ [23], while more peaks exist in the type B hydrate, including six major peaks of 2879, 2912, 2937, 2967, 2992, and 3019 cm⁻¹, and two minor peaks of 2866 and 2894 cm⁻¹. As a result, all the Raman spectra collected from TBAB SCH formation from different mass concentrations correspond to type A TBAB hydrates.

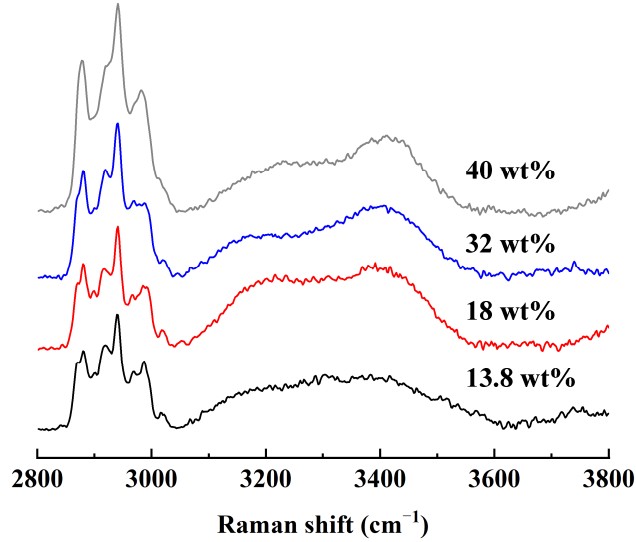

**Figure 3.** Comparison of Raman spectra of TBAB SCH formation from TBAB solution with mass concentrations of 13.8 wt%, 18 wt%, 32 wt%, and 40 wt%.

The equilibrium temperature of type A TBAB hydrates at these mass concentrations are 6.3 °C, 8.5 °C, 11.5 °C, and 12.1 °C, resulting in the degree of supercooling required for hydrate nucleation being 12.5 °C, 14.5 °C, 17.5 °C, and 17.9 °C for the 13.8 wt%, 18 wt%, 32 wt%, and 40 wt% solutions [24], respectively. The results align with the measurements performed using micro-differential scanning calorimetry (micro-DSC), which reported a supercooling temperature of (17.7 ± 0.7) °C for TBAB SCH with a 40 wt% concentration [21]. It is worth noting that in these experiments, the temperature decreased to 258.2 °C at a cooling rate of 0.5 °C/min, which coincides with our experimental conditions. The supercooling degree required for hydrate nucleation is an important parameter in practical application; a larger degree of supercooling consumes more energy. It can be found out that the lower the concentration of the solution, the less supercooling required for nucleation.

3.1.2. Optical Morphology

Figures 4 and 5 illustrates the optical morphology of TBAB SCH formation from TBAB solutions with different mass concentrations. It is evident from these figures that the morphology of TBAB SCH hydrates varies with the mass concentration of the TBAB solution. In the TBAB solution with a low mass concentration of 13.8 wt%, dendric shape of crystals are formed. Nucleation and growth processes rely on the mass transfer and heat transfer to develop the crystal. At lower mass concentration of TBAB solution, the formation of SCH involves mass condensation, which reduces the diffusion rate and

creates a diffusion field that destabilizes the growing surfaces. The dendric shape both facilitates the mass transfer and heat transfer process, because the perks and valleys increase the total interfacial area between the hydrate and water [25].

As shown in Figure 4d–f, more densely dendric crystals formed at mass concentration of 18 wt%. This is because of the more abundant mass supply in this solution. Further increase the mass concentration of the TBAB solution, there are no peaks and valleys along the crystal growth axis since the mass supply is sufficient as shown in Figure 5.

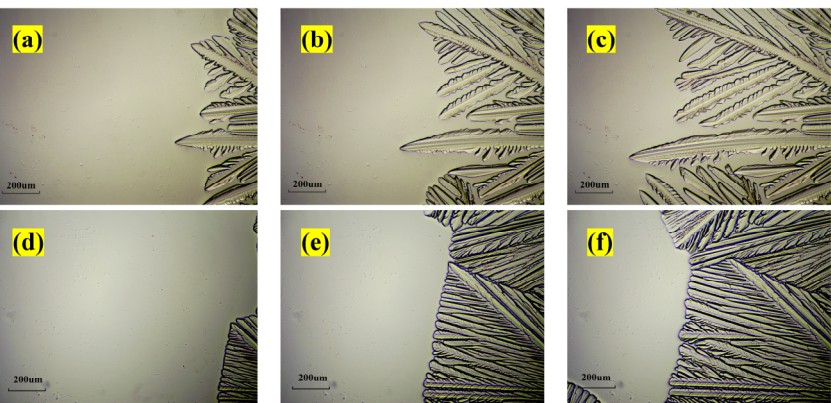

**Figure 4.** The optical morphology of TBAB SCH from TBAB solution with different mass concentrations: (**a**–**c**) 13.8 wt% and (**d**–**f**) 18 wt%.

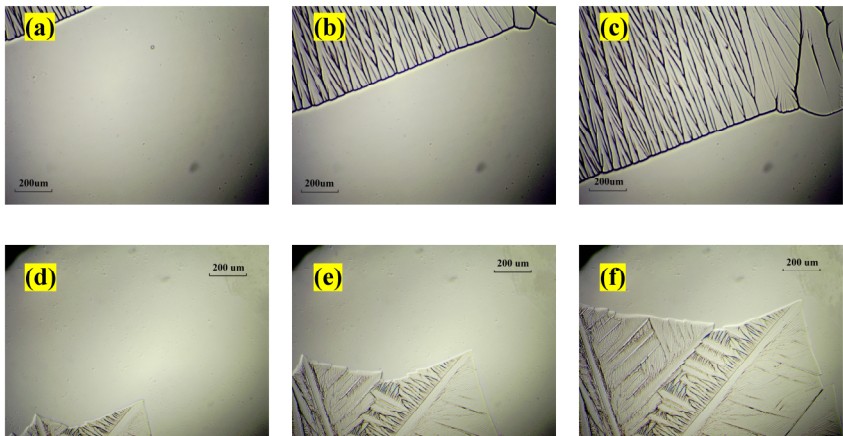

**Figure 5.** The optical morphology of TBAB SCH from TBAB solution with different mass concentrations: (**a**–**c**) 32 wt% and (**d**–**f**) 40 wt%.

### 3.2. Hydrate Formation with Different Cooling Rates

It has been reported that the cooling rate of the reactor can affect the crystal morphology and formation speed of the gas hydrates[26]. We carried out experiments to lower the temperature of the reactor from 13 °C to −10 °C at a cooling rate of 2 °C/min with TBAB solution of 13.8 wt%, 18 wt%, 32 wt% and 40 wt%.

In the case of the TBAB solution of 13.8 wt%, similar dendric shape of crystals form under larger cooling rate as shown in Figure 6a–c. Compared to the lower cooling rate in Figure 4a–c, the distance between the branches of the dendric crystal is larger because the cooling is very fast and the diffusion of TBAB particles is relatively slower. When the cooling rate increases, the crystal shape remains dendric type in TBAB solution of 18 wt% while with loosely growth pattern, same with the case in solution of 13.8 wt%.

With increases in the cooling rate, the TBAB SCH in the 32 wt% solution exhibits a snowflake-like morphology shown in Figure 7a–c. The snow crystals grow and come into

contact with other crystals; they connect with each other at irregular orientations. Subsequently, the connected crystals expand outwards, resembling clouds surging in the sky. Under circumstances of 40 wt% solution, crystals of hourglass shape first appeared and they evolve into swallowtail crystals with dendric fibers [27] developed along different directions in Figure 7d–f. This is because in dilute solution phases, the condensation process of mass transfer plays a crucial role in crystal growth. While in the case of denser solutions, heat transfer is the dominant factor [28].

Two types of crystal growth pattern are distinguished in this study: mass transfer-controlled growth (skeletal crystals) and growth by diffuse heat control (snow crystals). This is consistent with the observation of olivine morphology at different cooling rates and degrees of supercooling [29].

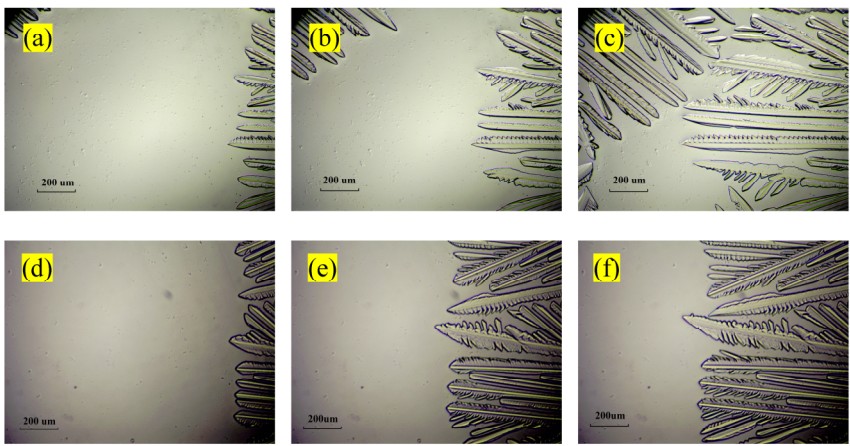

**Figure 6.** The optical morphology of TBAB SCH from TBAB solution with different mass concentrations: (**a–c**) 13.8 wt%; (**d–f**) 18 wt% at higher cooling rate of 2 °C/min.

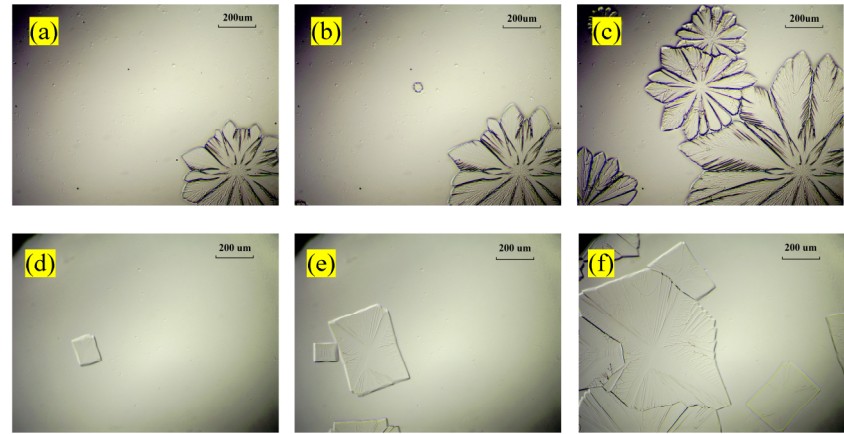

**Figure 7.** The optical morphology of TBAB SCH from TBAB solution with different mass concentrations: (**a–c**) 32 wt%; (**d–f**) 40 wt% at a higher cooling rate of 2 °C/min.

We also discovered that the hydrate formation temperature of the TBAB SCH increased with the increasing cooling rate. Figure 8 shows the nucleation temperature change with the cooling rate at larger mass concentration of 32 wt% and 40 wt%. As solution of 32 wt%, the nucleation temperature is similar at approximately −6 °C when the cooling rate is 0.5 °C/min and 2 °C/min, while further increasing the cooling rate to 4 °C/min and 6 °C/min, the nucleation temperature increases to −3.7 °C and 1.2 °C, even above the ice point. In case of 40 wt%, the nucleation temperature nearly increases with the cooling rate and it is 0.3 °C at a cooling rate of 6 °C/min. This is because a large

temperature gradient appears in a short time, which reduces the Gibbs free energy barrier for hydrate formation, and the hydrate will rapidly nucleate in a short time [30].

At the dilute solutions of 13.8 wt% and 18 wt%, when the cooling rate increases to 4 °C/min and 6 °C/min, we found that there appears the "competition" of the ice formation from water and TBAB hydrate formation. As explained previously, with increase of the cooling rate, the Gibbs free energy barrier for hydrate formation and ice formation both decreases. Thus, the nucleation temperature in the dilute solutions is not shown in the figure since it is not illustrative for TBAB-based applications. While in practical applications, the cooling rate should be carefully considered in order to avoid ice formation that hinder TBAB hydrate formation.

This finding is significant for practical applications, as excessive supercooling can lead to issues such as unreliable heat storage and increased cooling costs. Increasing the cooling rate is an effective approach to induce large temperature gradients, and, thus the degree of supercooling that required for hydrate nucleation, which can facilitate the practical implementation of TBAB SCHs in various applications.

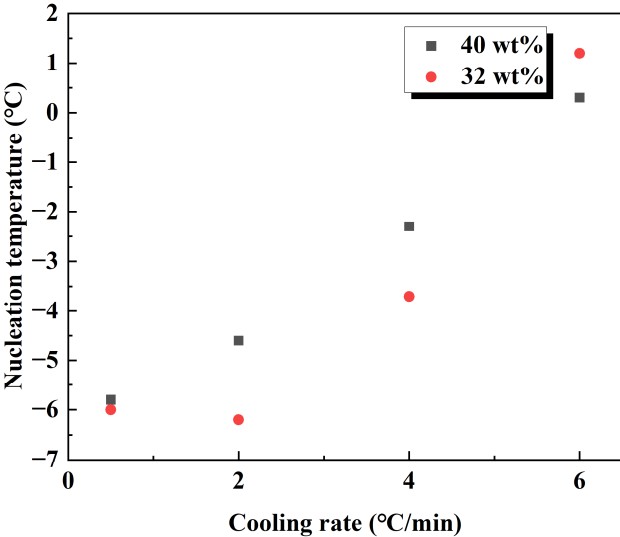

**Figure 8.** The hydrate nucleation temperature variation with the cooling rate of the reactor at a TBAB mass concentration of 32 wt%.

## 4. Discussion

From a thermodynamic perspective, type A hydrates show better stability at mass concentrations larger than 20 wt%, while type B hydrates are more stable in dilute solution conditions [24]. However, in our experiment, a type A TBAB hydrate was preferentially formed in all TBAB solutions with varying mass concentrations.

Shi et al. investigated the crystallization behavior of TBAB hydrates with 15 wt% solutions and found that a large degree of supercooling was not beneficial for type B hydrate formation [31]. When the reactor was cooled to 4.4 °C, type A preferentially formed and had a certain probability to transform into type B hydrates. Kim et al. also observed that type A preferentially formed at a TBAB solution of 20 wt% [23]. The same group conducted another work related to the growth behavior of TBAB hydrate from a 10 wt% solution and also discovered the preferential nucleation of the type A hydrate [32]. From their perspective, the local water network structure between type A hydrates and TBAB solutions shows more similarity than that between type B and TBAB solution. Chazallon et al. estimated the ratio of pentagons to hexagons of type A and B hydrates to predict the vibrational behavior of the host lattice [22]. The ratio of type A was smaller than that of type B; thus, the host lattice O-H vibrational spectra of type A show "water-like" behavior.

Nevertheless, the explanation for this phenomenon remains unclear. As is known, gas hydrate formation involves two steps: the formation of a crystallization nucleus followed by the subsequent growth of hydrate crystal after the nucleus reaches the critical nucleation size [33,34]. Li et al. calculated the maximum critical nucleation size of different concentrations of TBAB solution with the classical nucleation theory [30]. The critical size for TBAB hydrate nucleation in 10 wt% was 15.6 Å (±6), lower than that in a higher-concentration solution of 30 wt%, which was 29 Å (±6). The critical nucleation size of type A TBAB SCHs may be smaller than that of type B hydrates, making it kinetically favored, despite being thermodynamically unfavorable under certain circumstances. Further molecular-level investigations may shed new light on the mechanism underlying this phenomenon.

## 5. Conclusions

This work is focused on the morphological study of TBAB SCHs in confined spaces with an optical microscope and Raman spectroscopy. Many parameters directly affect the crystal morphology [35], with the mass concentration of the TBAB solution and the cooling rate being investigated in this study. Although type B TBAB SCH is thermodynamically stable in dilute TBAB solutions, type A TBAB SCH is preferentially formed in all of the TBAB solutions with concentrations of 13.8 wt%, 18 wt%, 32 wt%, and 40 wt%. This may be attributed to the smaller critical nucleation size of type A hydrates, while further research on the molecular level should be conducted to fully explain this phenomenon. At a larger cooling rate of 2 °C/min, two distinct crystal growth patterns are distinguished, including mass transfer-controlled growth and growth regulated by heat transfer. Increasing the cooling rate not only changes the optical morphology, but also reduces the supercooling temperature due to a decrease in the Gibbs free energy barrier caused by the increased temperature gradient. This finding has practical implications for TBAB applications, as it helps to mitigate the degree of supercooling.

**Author Contributions:** Conceptualization, L.G. and H.L.; methodology, L.G.; writing—original draft preparation, L.G.; writing—review and editing, L.G. and H.L.; visualization, L.G.; supervision, H.L.; project administration, H.L.; funding acquisition, H.L. All authors have read and agreed to the published version of the manuscript.

**Funding:** This work was supported by the China Geological Survey [grant number DD20230063 and DD20221073] and the Guangdong Major project of Basic and Applied Basic Research [grant number 2020B0301030003].

**Data Availability Statement:** Data are contained within the article.

**Acknowledgments:** We thank Wenjiu Cai and Lei Wang for the technical support and useful discussion about the experiments. We thank Yi Zhang for the funding support.

**Conflicts of Interest:** The research was conducted in the absence of any commercial or financial relationships that could be construed as a potential conflict of interest.

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
