# Peer review of "Morphological Study of Tetra-n-Butylammonium Bromide Semi-Clathrate Hydrate in Confined Space"

_crystals, doi:10.3390/cryst14050408_

Round 1
Reviewer 1 Report
Comments and Suggestions for Authors
The paper “Morphological study of Teta-n-butylammonium bromide semi-clathrate hydrade in confined space” shows pictures of different mixtures of Teta-n-butylammonium bromide with water nucleation.
I believe that this paper needs a substantial expansion and can be published only after major revision.
1. I personally find the experimental results presentation quite poor. First of all, the Raman studies are conducted for all 4 mixtures, while optical morphology is studied for 3 of them, dependence on cooling rate – for 2, and finally, only for one, 32 wt%. But the manuscript does not explain why. And this I find very important to explain.
2. Also, if the only results are pictures of grown hydrates, I find that there are not enough of them to make any conclusions. Please, present pictures of the just-nucleated hydrates, different forms that can be found. Widen the criteria of study.
3. Also, there already are a number of numerical and descriptive techniques for assessing morphology specifically, for example fractal and multifractal analysis, Minkowski functionals, etc. I feel that in modern science simple statement that some hydrates grow in the form of trees and others are like a line is not enough.
I found several small mistakes and typos.
4. I doubt about the reference [2]. It does describe carbon dioxide capture, but the paper does not contain any mention of SCH. Probably it might be incorrect to cite this reference in this paper.
5. Dimensions of the optical sheet (Fig 1b) should be in mm, mkm, or whatever is applicable. Either authors should add this to the figure itself, or to the figure caption.
6. Line 121 – It should be “peaks”, not “perks”.
Line 181 - It should be “cooling rate”, not “cooling late”.
Comments on the Quality of English LanguageSmall mistakes and typos were detected. Can be easily fixed.
Author Response
Thanks for your useful suggestions and please see my response in the attachment.

Reviewer 2 Report
Comments and Suggestions for Authors
The manuscript on the 'Morphological study of Tetra-n-butylammonium Bromide Semi-clathrate Hydrate in confined space´ describes the effect of two factors, temperature and concentration, in obtaining morphologies and proportions of two crystalline phases (A and B), using an in situ morphological observation device, through a polarising microscope, in a confined space made with an optical sheet etched with a
a 0.03 mm deep groove. The optical foil is placed over a reactor capable of withstanding high pressure up to 5 MPa and low temperature down to -110 ℃.
Although the experiment is interesting, and well performed, I find that some things should be better explained or substantiated:
- The experiment is controlled by polarisation microscopy and Raman spectroscopy only. A crystallographic characterisation at the molecular level of phases A and B is missing. Is it possible to quantify the crystalline compositions from the Raman spectra?
The crystal structures, real or modeled, of the two phases should be described in the manuscript, to facilitate the reading.
I wonder whether a powder diffraction experiment could be performed to give a quantitative approach to the actual crystalline composition of the different experiments.
Both the conclusions and the discussion are very general, a more quantitative and not only phenomenological discussion should be made, based on the experimental data and considering the limitations of this study.
Should this concerns being resolved and the manuscript revised accordingly, the manuscript may be consider for publication.
Comments on the Quality of English LanguageThe paper is readable and have no special text issues.
Author Response

(The authors gave the same response as above.)

Round 2
Reviewer 1 Report
Comments and Suggestions for Authors
The authors have made a lot of work. Now the paper can be published
Reviewer 2 Report
Comments and Suggestions for Authors
As my concerns have been satisfactorily answered and the manuscript have been revised, the manuscript can be now accepted.